# Three-Dimensional Microtumor Formation of Infantile Hemangioma-Derived Endothelial Cells for Mechanistic Exploration and Drug Screening

**DOI:** 10.3390/ph15111393

**Published:** 2022-11-12

**Authors:** Yanan Li, Xinglong Zhu, Meng Kong, Siyuan Chen, Ji Bao, Yi Ji

**Affiliations:** 1Division of Oncology, Department of Pediatric Surgery, West China Hospital of Sichuan University, Chengdu 610041, China; 2Med-X Center for Informatics, Sichuan University, Chengdu 610041, China; 3Institute of Clinical Pathology, Key Laboratory of Transplant Engineering and Immunology, NHC, West China Hospital, Sichuan University, Chengdu 610041, China; 4Pediatric Intensive Care Unit, Department of Critical Care Medicine, West China Hospital of Sichuan University, Chengdu 610041, China

**Keywords:** drug screening, infantile hemangioma, three-dimensional, microtumor model, extracellular matrix

## Abstract

Infantile hemangioma (IH) is the most prevalent type of vascular tumor in infants. The pathophysiology of IH is unknown. The tissue structure and physiology of two-dimensional cell cultures differ greatly from those in vivo, and spontaneous regression often occurs during tumor formation in nude mice and has severely limited research into the pathogenesis and development of IH. By decellularizing porcine aorta, we attempted to obtain vascular-specific extracellular matrix as the bioink for fabricating micropattern arrays of varying diameters via microcontact printing. We then constructed IH-derived CD31+ hemangioma endothelial cell three-dimensional microtumor models. The vascular-specific and decellularized extracellular matrix was suitable for the growth of infantile hemangioma-derived endothelial cells. The KEGG signaling pathway analysis revealed enrichment primarily in stem cell pluripotency, RAS, and PI3KAkt compared to the two-dimensional cell model according to RNA sequencing. Propranolol, the first-line medication for IH, was also used to test the model’s applicability. We also found that metformin had some impact on the condition. The three-dimensional microtumor models of CD31+ hemangioma endothelial cells were more robust and efficient experimental models for IH mechanistic exploration and drug screening.

## 1. Introduction

Infantile hemangioma (IH) is the most frequent childhood vascular tumor; it is a noncongenital benign neoplasm that affects approximately 4% to 5% of newborns [1]. The incidence of IH appears to have increased significantly over the last 40 years, which could be associated with an increase in low birth weights and prematurity [2]. Although IH is typically small and regresses naturally in many cases, it can be extremely disfiguring and can obstruct airways, impair vision, and cause congestive heart failure in some cases [3].

The pathogenesis of IH is still unclear and is believed to be derived from the embryo [4]. Hypoxia and angiotensin, as independent factors, have been suggested to play an important role in the development of IH with a synergistic effect [5,6]. Recently, some evidence has suggested that the renin–angiotensin system (RAS) is the driving force in IH development, and the vasoactive peptide angiotensin II has been speculated to promote the secretion of osteoprotegerin and the VEGF system, thereby acting as a protumor factor for vasculogenesis and an antiapoptotic factor [7,8,9]. In 2008, Leaute-Labreze et al. revealed hemangioma regression in infants receiving propranolol for cardiac disease [10]. Propranolol, a nonselective receptor blocker, has become the first-line treatment for IH. However, the mechanism of action of propranolol on IH remains unknown [11]. Propranolol is considered safe and effective in adults, but its safety in infants has not been well established, with a variety of adverse reactions leading to treatment discontinuation and IH recurrence [11]. Additionally, the patients in the initial group that was treated with propranolol are still in puberty, and the long-term neurodevelopmental implications are still unknown [11]. As a result, it is critical to develop safer and more effective medications.

Presently, due to the lack of stable cell lines, primary cells (primarily including hemangioma-derived endothelial cells (HemECs) and hemangioma-derived endothelial progenitor cells (HemEPCs)) must be extracted for fundamental research on IH [12]. There are significant structural and physiological differences between two-dimensional (2D) cell cultures and those in vivo. In addition, spontaneous regression frequently occurs during neovascularization in nude mice, severely limiting studies on the pathophysiology and development of IH [1,13]. In vitro, three-dimensional (3D) tumor cell growth has been shown to greatly increase tumor cell function, histomorphology, viability, stability of genotype, and pharmaceutical metabolism. Their cellular aggregates are coated with natural extracellular matrix (ECM), which in 3D environments more closely mimics those of in vivo tumors [14,15]. Recently, some scholars have used covalent and noncovalently coated carbohydrates, peptides, and proteins to design a microarray in a petri dish and determine the shape and size of the pattern formed on the base. In this model, cells are restrictively adhered to the micropattern and can spontaneously assemble into a 3D structure of multicellular spheroids via cell proliferation and cell-cell adhesion [16]. Thus, it is possible to develop 3D multicellular spheroids with a controlled size and an ordered layout, which is beneficial for high-throughput drug screening. Such tumor spheroids have been referred to as “microtumors” or “microcancers” in some investigations [14,17]. With microcontact printing technology, proteins (as bioinks) adsorbed on polydimethylsiloxane (PDMS) form protein-specific microarrays on unconnected petri dishes. The fields of cytology, drug screening, and tissue engineering have all made extensive use of this technique. Bioinks such as collagen I, fibronectin, laminin, and collagen IV are frequently employed in traditional micropattern arrays [18]. However, the single ECM protein component is insufficient to regulate and enhance the function and viability of certain primary cells, cell lines, or differentiated cells produced from iPS cells [19]. Recent breakthroughs in the decellularization of organs and tissues have made it feasible to obtain an ECM with a distinct structure and biological and organ specificity [19]. For instance, liver-specific ECMs containing a variety of biological macromolecules can regulate signaling pathways involved in cell proliferation, migration, and differentiation by interacting with tumor cells via disc-domain receptors and transmembrane proteoglycans or integrins, thereby influencing cell biological behavior [20].

Mulliken’s groundbreaking discovery demonstrated that HemECs could be isolated; their characteristics subsequently were studied in vitro [21]. From the proliferating hemangioma specimens, they extracted an immature progenitor-like cell, which Joyce described [22]. These researchers referred to the cells as stem cells originating from hemangiomas (HemSCs), and the model was effectively established in nude mice. However, Joyce revealed that one limitation of their research was that the mouse hemangioma lesions did not significantly expand as one might anticipate from a proliferating hemangioma [22]. Huamin Mai showed that HemSCs were also exceptional in their ability to be cloned from a single cell to a population of up to one billion cells with strong proliferation, production of GLUT-1 (an IH-specific marker)-positive microvessels, and transformation into adipocytes at subsequent time periods, replicating the typical progression of IH [23]. The investigators also discovered that the proliferation of HemSCs in mice was not quick or plentiful, similar to that of component cells in proliferating IH, and insufficient blood was available to conduct a pharmacological test on the Matrigel-formed vasculature [23]. Moreover, for physiology and disease modeling, animal models are constrained by their higher cost, ethical dilemmas, and xenogeneic nature [24]. Drugs are typically tested in 2D in vitro cell cultures, but due to the greatly diminished cell-cell and ECM interactions, this method is not able to accurately replicate the natural characteristics of the in vivo tumor microenvironment.

In fact, the lack of an effective in vivo IH model and problems in obtaining IH patient samples before and after treatment have hampered efforts to uncover propranolol’s mechanism of action [1]. Many attempts to construct an in vivo IH model have failed to demonstrate propranolol’s antitumor effect [1,25]. Moreover, the pathophysiology of IH is currently unknown, and standardized and robust research models are desperately needed. We successfully separated and cultured primary IH endothelial cells in a previous study and obtained CD31+ hemangioma endothelial cells (CD31+ HemECs) via flow cytometry [26]. In this study, a high-throughput CD31+ HemEC 3D microtumor model with a uniform size was constructed using vascular-specific and decellularized ECM (dECM) and a micropattern array to provide a more robust and efficient experimental model for IH mechanistic exploration and drug screening.

## 2. Results

### 2.1. Characterization of DPHASs

The DPHASs were obtained by decellularizing porcine heart aortic tissue according to the study protocol, and their properties were assessed. The DPHA tissue was slightly smaller but preserved its original form. H&E staining revealed no evident nuclear or cytoplasmic residues in the DPHASs, and ECM fibers were retained (Figure 1A). Compared to normal tissues, the DPHASs lacked cellular components but preserved a dense network of ECM collagen fibers, as revealed by Masson’s trichrome (MT) staining (Figure 1B). Immunohistochemical labeling revealed that all cells were eliminated, whereas in the ECM of the DPHASs, the collagen Ⅰ, collagen Ⅲ, and collagen Ⅳ were mostly preserved compared to those of normal tissues (Figure 1D–F). The dry weight ratio of the DNA content in the normal tissues was 663.5 ± 17.9 ng/mg, whereas the DPHASs had a dry weight ratio of just 29.2 ± 3.3 ng/mg.

### 2.2. Evaluation of dECM-Coated Plate Characteristics

These results showed the applicability of the dECM-coated plate for the culture of CD31+ HemECs. The DPHA was ground into a powder after being lyophilized. The DPHA powder was then broken down by pepsin to generate the dECM hydrogel. For analysis of the attachment capacity, each well in coated and untreated 24-well plates was seeded with 3 × 10^5^ CD31+ HemECs for 10 min. The dECM-coated plates had a greater adhesion rate than the untreated, collagen I-coated, or Matrigel-coated plates of up to 62.2% (Figure 2B). Each well of the treated and untreated 96-well plates was seeded with 5 × 10^3^ HepG2 cells and cultivated for 24 h. After incubation of the CCK-8 working solution for 2 h, a microplate reader was used to calculate the OD value. Compared to the untreated and collagen I-coated plates, the dECM-coated and Matrigel-coated plates exhibited superior cell proliferation rates (Figure 2C). Each well of the treated and untreated 24-well plates was seeded with 2.5 × 10^4^ CD31+ HemECs. On days 1, 3, and 5 of cell culturing, fluorescence staining was carried out. On the first day of culturing, the dECM-coated and Matrigel-coated plates had a higher cell density than the untreated and collagen I-coated plates (Figure 2A and Appendix A). On the third and fifth days of culturing, the morphology of the CD31+ HemECs from the dECM-coated and Matrigel-coated plates was more homogeneous than that on the untreated and collagen I-coated plates (Figure 2A).

### 2.3. Specific Features of the Micropattern Array

The characteristics of microtumors with various-sized micropattern arrays were evaluated. The PDMS seals reliably produced 50, 100, 150, and 200 μm circular micropattern arrays on dishes containing dECM hydrogel in bright field and fluorescent light (Figure 3A,B). The CD31+ HemECs was seeded onto the micropattern array covered with dECM. The monolayer CD31+ HemECs was attached to each micropatterned site after 6 h, and nonadherent cells were rinsed with PBS (Figure 3C). Consistent and orderly cell microtumors could be observed after three days of culturing (Figure 3D). The average diameter of the microtumors was 58.2 ± 5.4, 103.1 ± 4.2, 161.8 ± 4.7, and 205.8 ± 5.1 μm, respectively. The FluoroQuench results revealed that the cell viability was 87.3 ± 2.7%, 90.9 ± 2.6%, 83.5 ± 1.7%, and 80.9 ± 4.6% for the 50, 100, 150 and 200 μm patterns, respectively (Figure 3E). On the 50 and 100 μm circular micropatterns, cell spheres developed with a spherical shape; on the 150 and 200 μm circular micropatterns, cell spheroids formed with a hemispherical shape (Figure 4A–D). The expressions of two vascular endothelial-specific genes (MMP-2 and VEGF-A) were quantified using RT-PCR. MMP-2 and VEGF-A cell mRNA levels were substantially greater than those of the monolayer cells (*p* < 0.05). MMP-2 and VEGF-A were the most highly expressed in the 100 μm micropatterns followed by the 150 μm pattern (Figure 4E). The above-mentioned findings revealed that the ideal diameter of the round patterns was 100–150 μm.

### 2.4. Results of the RNA Sequencing

These findings revealed transcriptional differences between CD31+ HemEC microtumors and 2D cell models. Transcriptome sequencing was used to investigate the process by which CD31+ HemECs self-organized into microtumors. Compared to those in the 2D cell model, 1288 genes were significantly upregulated, and 931 genes were significantly downregulated (Figure 5B). The GO enrichment analysis for significantly upregulated genes mostly identified single-organism processes, regulation of cellular processes, protein binding, and so on (Figure 5D). The KEGG signaling pathway enrichment analysis showed enrichment mainly in the signaling pathways that regulate the pluripotency of stem cells, the RAS signaling pathway, and the PI3K-Akt signaling pathway (Figure 5E). Focal adhesion-related genes, including SHC4, COMP, PIK3R3, KDR, and PAK3, were significantly upregulated as verified by qRT–PCR (Figure 5C).

### 2.5. Microtumor Characteristics

The characteristics of microtumors generated on 100 μm diameter micropatterns were evaluated using immunocytochemistry and fluorescence probe staining. The CD31+ HEC microtumors had increased expressions of specific immunity indicators such as GLUT-1, proliferative markers such as Ki67, hypoxia markers such as HIF-1ɑ, and antiapoptotic markers such as MCL-1 compared to the monolayer cells, as revealed by immunofluorescence images (Figure 6A–D). Interestingly, cells positive for MCL-1 and HIF-1ɑ were identified in the center of the microtumors, whereas cells positive for Ki-67 were distributed around the microtumor perimeter (Figure 6B–D). The cytosolic pH of the microtumors was lower than that of the monolayer cells as evidenced by the higher fluorescence intensity of the pH indicator (Protonex Red) in the microtumors than in the monolayer cells (Figure 6F). Additionally, the expression of N-cadherin was widely concentrated in places where cells interacted with one another, which was indicative of the formation of basal–lateral polarity (Figure 6E).

### 2.6. Drug Screening

Propranolol and metformin were used to assess the model’s suitability for drug screening. On 100 μm micropattern arrays, CD31+ HemEC microtumors with a good cellular viability were generated after 3 days. The CD31+ HemEC microtumors were treated with propranolol (50 μM, 100 μM, 150 μM, and 200 μM) and metformin (5 mM, 10 mM, 15 mM, and 20 mM) for 24 and 48 h. As a control, 0.25% DMSO was employed. The cell viability rate was approximately 52.9% when the propranolol dosage was 100 μM at the 2D culture level according to the CCK-8 analysis (Appendix A). The cell viabilities of propranolol were 71.3 ± 4.5%, 64.7 ± 2.3%, 54.1 ± 1.6%, and 46.7 ± 2.3%, respectively, at 50 μM, 100 μM, 150 μM, and 200 μM at 24 h; while the cell viabilities were 62.4 ± 0.5%, 54.2% ± 0.4, 49.7 ± 1.5%, and 39.3 ± 1.2%, respectively, at 48 h after drug administration (Figure 7A). The cytotoxic effect of metformin was low; the cell viabilities of metformin were 72.1 ± 1.1%, 61.3 ± 4.4, 50.6% ± 0.8%, and 45.0 ± 1.9%, respectively, at 5 mM, 10 mM, 15 mM, and 20 mM at 24 h; while the cell viabilities were 73.1 ± 2.8%, 55.7 ± 0.6%, 45.5 ± 2.8, and 40.8 ± 2.0%, respectively, at 48 h after drug administration (Figure 7B).

## 3. Materials and Methods

### 3.1. Decellularization of Porcine Heart Aortic Tissue

Male Bama tiny pigs (measuring 30–40 kg) were provided by Sainuo Biomedical (Chengdu, China). All animal protocols were carried out at the Sichuan University Research Center with the approval of the Sichuan University Animal Experiment Center (item number: 20220307040) and consistent with the Laboratory Animal Welfare Act and standard operating procedures. The fresh porcine heart aortic tissue was immediately submerged in phosphate-buffered saline (PBS) that contained 100 g/mL streptomycin and 100 U/mL penicillin on ice after being harvested from the pig. For lysis of the blood cells, the aorta was rinsed with distilled water on an orbital shaker (100 r/min) for 24 h. The sample was then immersed in 0.5% Triton X-100 solution and oscillated for 24 h (at 100 r/min). The samples were immersed in 0.25% SDS solution and oscillated continuously for 72 h (100 r/min) after being washed in PBS for 2 h. For elimination of detergent residue, the sample was oscillated for 72 h (100 r/min) with PBS. The porcine heart aortic tissue in the control group was only rinsed with distilled water.

Paraformaldehyde-fixed native or decellularized porcine heart aortic tissues were embedded in paraffin and sectioned for hematoxylin and eosin (H&E) staining and Masson’s trichrome (MT staining) (G1340, Solarbio, Beijing, China). The images were taken with a NanoZoomer Digital Pathology scanning system (Hamamatsu, Hamamatsu City, Japan). Furthermore, the preservation of important proteins in the porcine heart aortic tissue was assessed using primary antibodies against collagen Ⅰ (ab233080, 1:100, Abcam), collagen Ⅲ (ab6328, 1:200, Abcam), and collagen Ⅳ (ab6586, 1:500, Abcam). The total DNA was removed from five independent decellularized and native porcine heart aortic tissues (10 mg dry weight) using a Tissue DNeasy Kit (Tiangen Biotech, Beijing, China), and measured using a NanoDrop One spectrophotometer (Thermo, Waltham, MA, USA).

### 3.2. Preparation of the Vascular-Specific dECM Hydrogel

For lyophilization, the decellularized porcine heart aortic (DPHA) tissues were cut into 1 cm × 1 cm × 1 cm disks, and the lyophilized DPHA scaffolds (DPHASs) were ground into a powder using a Wiley mill (Retsch, MM400, Haan, Germany). An 80-mesh screen was used to filter the ground scaffolds. For generation of the dECM hydrogel, the ground scaffolds were agitated for 48 h with 10% pepsin and 0.01 M HCl at ambient temperature. The dECM hydrogel was then neutralized with 0.1 M NaOH to a pH of 7.2–7.4. The final concentration of the dECM hydrogel was adjusted to 10 mg/mL with 1 × PBS and stored at 4 °C.

### 3.3. Cell Culture

The CD31+ HemECs’ extraction and isolation were performed as previously described with the approval of the human ethics review committee [26,27]. The CD31+ HemECs were cultivated in endothelial basal medium (EBM-2, Lonza, Walkersville, MD, USA), which included a 1% penicillin–streptomycin solution (HyClone, Logan, UT, USA) and a 10% fetal bovine serum (Gibco, New York, NY, USA) according to our previous research. The cells were cultured in a humidified environment that contained 5% CO_2_ while maintaining a temperature of 37 °C.

### 3.4. Evaluation of Plate Coating and Features

PBS was used to adjust the concentration of the dECM hydrogel to 0.1 g/L. Collagen I (BD Biosciences, Bedford, MA, USA) and Matrigel (Corning, New York, NY, USA) with the same concentration of 0.1 g/L were used as the positive coating controls, while untreated plates served as the negative controls. After addition of 500 microliters of coated hydrogel to 24-well cell culture plates, the plates were incubated in an incubator for one hour at 37 °C. The extra liquid was removed, and the plates were washed with PBS until they were clean. The CD31+ HemECs were seeded onto the plates and cultured at 37 °C as described above. A Countess^TM^ II FL automated cell counter (Invitrogen, Carlsbad, CA, USA) was used to count nonadherent cells after 2 h of culture. A fluorescence microscope (Axio Observer D1/AX10 cam HRC, Carl Zeiss, Gottingen, Germany) was used to image the representative materials. A Cell Counting Kit-8 (CCK-8, MCE, Monmouth Junction, NJ, USA) assay was used to estimate cell proliferation.

### 3.5. Printing of Micropattern Arrays

As revealed in our previous study, polydimethylsiloxane (PDMS) seals were created utilizing a silicon wafer with a laser-etched characteristic pattern [28]. The PDMS seals were fabricated by Suzhou CChip Scientific Instrument Co., Ltd., Suzhou, China. A 20 mm × 20 mm PDMS substrate was used to form the round designs with 50, 100, 150, and 200 μm diameters and 50 μm interspaces. The PDMS seals were covered with 0.1 g/L of filtered and sterilized dECM hydrogel for 20 min at room temperature. After removal of the extra dECM hydrogel, the PDMS seals were dried for 10 min at 37 °C. Next, the seals were subjected to 0.2 N pressure for 10 min while placed on a 35 mm nontreated cell culture dish. The FITC-labeled microarray morphology was examined using a fluorescence microscope. For future usage, the processed dishes were maintained at room temperature and away from light. Before seeding the cells, the processed dishes were first treated with a 10 g/L pluronic F-127 solution (Sigma–Aldrich, St. Louis, MO, USA) for one hour to inhibit nonspecific cellular adhesion, after which they were sterilized using ultraviolet irradiation for one hour.

### 3.6. Culturing of CD31+ HemEC Microtumors

In the dECM-printed dishes, 3 mL of medium was added, and CD31+ HemECs (2 × 10^5^) were inoculated. The medium was withdrawn after 6 h of incubation, and the nonadherent cells were washed with PBS three times. The culture was continued by addition of 2 mL of medium and daily medium changes. A control group consisting of monolayer cells that were cultivated under the identical circumstances as microtumors was used in this experiment. The shape of the microtumors was evaluated every day with an EVOS^TM^ XL Core (Invitrogen). ImageJ software was utilized to measure and analyze the diameters of the microtumors. Following the manufacturer’s instructions, a FluoroQuench fluorescent viability stain was utilized to evaluate the viability of the microtumors. Confocal microscopy (Nikon) was used to image the representative samples.

### 3.7. RNA Sequencing

RNA was extracted from six samples (3 samples of microtumors and 3 samples of monolayer cells) using TRIzol (Invitrogen, catalog number 15596026) according to the procedures by Chomczynski et al. [29]. After RNA extraction, DNase I was used to degrade the DNA, and the RNA quality was assessed by evaluating A260/A280 with a Nanodrop^TM^ One^C^ spectrophotometer (Thermo Fisher Scientific, Inc, Waltham, MA, USA). After confirmation of the RNA’s integrity using agarose gel electrophoresis at a 1.5% concentration, qualified RNAs were eventually quantified using Qubit 3.0 with the help of a QubitTM RNA Broad Range Assay kit (Life Technologies, Q10210).

For the production of the stranded RNA sequencing libraries, 2 g of total RNA was used; the Ribo-off rRNA Depletion Kit (Human/Mouse/Rat) (catalog no. MRZG12324, Illumina) and the KCTM Stranded mRNA Library Prep Kit for Illumina^®^ (catalog no. DR08402, Wuhan Seqhealth Co., Ltd., Wuhan, China) were used as directed by the manufacturers. Using Illumina’s NovaSeq 6000 equipped with the PE150 model, the library products ranging from 200 to 500 bps were enriched, quantified, and sequenced.

### 3.8. Quantitative Real-Time Polymerase Chain Reaction (qRT–PCR)

The CD31+ HemEC microtumors were treated with TRIzol reagent (catalog number 15596-026, Invitrogen) to extract the total RNA. The iScript cDNA Synthesis Kit (Bio-Rad, Hercules, CA, USA) was then used to manufacture the complementary DNA (cDNA) using 1 g of total RNA as the starting material. The housekeeping gene glyceraldehyde-3-phosphate dehydrogenase (GAPDH) was used as an endogenous internal control in this study. The gene expression was evaluated and quantified using the Stratagene analysis software and the 2-Ct approach after running PCRs in triplicate (for the promoter sequences, see Appendix A).

### 3.9. Features of CD31+ HemEC Microtumors

The features of the CD31+ HEC microtumors were further determined using immunocytochemistry and fluorescence probe staining. Primary antibodies were used to stain monolayer cells or microtumors that had been preserved with 4% neutral formalin. Cell proliferation of CD31+ HemEC microtumors was detected using Ki-67 (ab16667, 1:100, Abcam) immunocytochemistry. The hypoxic microenvironment of the CD31+ HEC microtumors was assessed using HIF-1α (ab51608, 1:100, Abcam) immunocytochemistry. MCL-1 (ab32087, 1:100, Abcam) immunocytochemistry revealed the antiapoptotic activity of CD31+ HemECs as well as the expression of GLUT-1 (ab115730, 1:100, Abcam), an IH marker, in the CD31+ HemEC microtumors. The monolayer cells or microtumors were additionally stained with N-cadherin (66219-1-Ig, 1:100, Proteintech, Chicago, IL, USA) to measure the cell adhesion and interaction.

Following the manufacturer’s instructions, the cytosolic pH was evaluated with the fluorescent dye probe Protonex^TM^ Red 600 (catalog no. 21,207, 1 M; AAT Bioquest). After incubation of the cells for 30 min at 37 °C, the cells were rinsed gently with Hanks solution containing 20 mM HEPES and evaluated.

### 3.10. Evaluation of Chemotherapeutic Medication Cytotoxicity

The CD31+ HemEC microtumors were treated with different doses of propranolol (50 μM, 100 μM, 150 μM, and 200 μM) and metformin (5 mM, 10 mM, 15 mM, and 20 mM) for 24 and 48 h. For the purpose of serving as a control, microtumors were exposed to 0.25% DMSO. The vitality of the CD31+ HemEC microtumors was used to assess the drug cytotoxicity. FluoroQuench fluorescence staining was then used to assess the vitality of the microtumors.

### 3.11. Data Analysis

The statistical analysis in this study was carried out using SPSS 21.0 (SPSS, Inc., Chicago, IL, USA). All quantitative data are provided as the mean ± SD. Dunnett’s test was used for the quantitative analysis. ANOVA was used for the multiple stats tests. Any *p*-values that were lower than 0.05 were considered statistically significant. R software version 4.1.0 was used for the RNA-Seq data analysis (https://www.R-project.org. accessed on 1 July 2021).

## 4. Discussion

We produced a dECM-based hydrogel synthesized from DPHA tissue and then utilized PDMS to generate micropattern arrays. We then successfully constructed a homogenous 3D IH-derived CD31+ HemEC model. Transcriptome sequencing of CD31+ HEC spheres revealed 1288 upregulated and 931 downregulated genes linked to pellet formation. The GO enrichment analysis focused on cellular processes for upregulated genes. The KEGG signaling pathway enrichment study indicated that it was mainly enriched in stem cell pluripotency, RAS, and PI3KAkt.We assessed the applicability of the model with propranolol, the first-line treatment for IH, and discovered that metformin had some effect on IH.

Microtumor multicellular architectures also closely resemble the structures of tumorous tissue, making them excellent in vitro tools for applications such as drug screening [15]. Previously, IH-derived cells were first cultured into microtumors using a nonadherent culture before being moved to an adherent matrix for further development [30,31]. However, this method cannot effectively control the shape and size of cell spheres, nor can it guarantee the model’s stability and repeatability; consequently, its application in drug screening and mechanistic research is limited. A tumor is more than just a collection of tumor cells; it also has ECM and stromal cells, which together form the tumor microenvironment (TME) [32]. The ECM is the predominant noncellular constituent of the tumor microenvironment [33]. The ECM is also dynamic and regulates the release of growth factors, cytokines, and chemokines, which have profound effects on cell proliferation, migration, and differentiation [34,35]. Overexpression and collagen deposition are common in the early stages of tumor development [36]. Tissue mechanical properties are improved by collagen deposition and crosslinking to alter biophysical cues and promote tumor development. The expression of organ-specific ECM proteins indicates that the development of tumors requires a particular matrix environment [37]. We found that using dECM-based hydrogels as the surface coating matrix improved cell adhesion, proliferation, survival, and phenotype expression in CD31+ HemECs. We constructed a high-throughput, homogenous CD31+ HemEC tumor sphere model by combining vascular-specific dECM with micropattern arrays. After examining micropattern arrays of varying diameters, we discovered that the appropriate diameter for cell morphology, viability, proliferation, and phenotypic expression was 100–150 μm.

The pathogenesis of IH is controversial and incompletely understood. The possibility of an embryonic origin was raised by the presence of placental markers in the endothelium of IH, including GLUT-1, the FC receptor, type 3 iodothyronine deiodinase, the Lewis-Y antigen, and merosin [38]. A variety of signaling pathways regulate the occurrence and development of hemangioma, such as regulating the pluripotency of stem cells, the RAS signaling pathway, and the PI3K-Akt signaling pathway [38,39,40]. In the present study, RNA sequencing was performed to investigate the mechanism by which CD31+ HemECs formed microtumors through self-organization; a KEGG signaling pathway enrichment analysis revealed that the pathway was enriched similarly to that previously reported. Moreover, SHC4, COMP, PIK3R3, KDR, and PAK3, which are associated with focal adhesion, were significantly upregulated. In 2007, SHC4 was discovered and positively identified in melanoma, and it was found to be critical in the development of metastatic melanoma in vivo [41]. It was further confirmed that SHC4 is specifically expressed at an early stage during the differentiation of embryonic stem cells and the development of embryos [42]. PIK3R3 is a phosphoinositide 3-kinase regulatory subunit that can result in the activation of the PI3K-AKT signaling pathway and has a remarkable impact on cell genesis, proliferation, differentiation, apoptosis, and metabolism [43,44]. According to previous reports, KDR is an essential factor in the pathogenesis of infantile hemangiomas [45,46]. Pak3, a serine/threonine kinase that is a member of the PAK family, plays crucial roles in numerous cellular processes, including cytoskeletal dynamics and cell motility [47]. Researchers have examined the role of COMP in the epithelial–mesenchymal transition (EMT) in the growth of tumors, but the precise regulatory mechanism is still unclear [48].

Hypoxia has been proposed as a potential independent pathogenic factor in the etiology of IH [5]. The increased expression of HIF-1α, along with its downstream targets (such as GLUT-1, VEGF-A, and insulin-like growth factor-2), and activating transcription factor 4 promote the survival of cells under hypoxic stress [5,6,49]. Circulating endothelial progenitor cells (EPCs) may come into contact with ischemic tissue and hypoxia-induced molecules (VEFG-A and HIF-1ɑ) that are essential to their growth and differentiation in tissues [50]. In CD31+ HEC microtumors, there were hypoxic core regions compared to those in the 2D culture model. We discovered that the tumor cells expressed HIF-1ɑ due to hypoxia in the interior of tumor cell microtumors in our work; high HIF-1ɑ expression was shown to be related to treatment resistance in several tumor cells [51]. The innermost cells of the tumor cell sphere are nonproliferating; hence, this region is known as the quiescent zone. The proliferating cells in the outer layer shield the cells in the quiescent zone. The quiescent zone’s cells can swiftly regain proliferative abilities when the outside cells die [33]. Ki-67-positive cells were predominantly dispersed in the outer layer of the CD31+ HemECs microtumors, while Ki-67-negative cells were located in the inner zone of the microtumors. The results showed that the cells in the outer layer of the microtumors were proliferative, whereas the cells in the inner zone were not.

The first-line treatment for IH is propranolol; more than 10 years have passed since this treatment was initially used [11]. Propranolol’s therapeutic effect is assumed to be related to its ability to affect endothelial cell (EC) apoptosis, vasoconstriction, and/or suppression of angiogenesis by modifying vascular endothelial growth factors. However, the precise mechanism of its action is unknown [1,52]. In fact, the lack of an in vivo IH model and the difficulty of obtaining samples from IH patients before and after treatment have both made it harder to discover the mechanism [1]. After 24 h of treatment in the 2D model of CD31+ HemECs, the 50% lethal concentration of propranolol was 100 μM. Interestingly, our research showed that the onset of clearly visible central necrosis within 24 h required a propranolol concentration of 150 μM. The results of our study showed that the CD31+ HemEC microtumors had good cell–cell adhesion, as demonstrated by N-cadherin immunocytochemistry. The cell-to-cell adhesion on the outside of the CD31+ HemEC microtumors formed a natural physical barrier, and the microtumors were generally very acidic. Hence, the CD31+ HemEC microtumor model showed a greater drug resistance than the 2D model. GLUT1 is a particular marker for IH that can be used to differentiate IH from other vascular lesions. Glycolysis is the main source of energy in tumor ECs, and it is further increased during angiogenesis [53,54]. In recent years, glycolysis-associated molecules such as GLUT-1, LDHA, and PFKFB3 have been found to have a higher expression in HemECs than in human umbilical vein ECs [55]. In our previous study, we discovered that inhibiting the glycolytic activator PFK-1 in HemECs could decrease the amount of glucose uptake as well as the production of lactate and ATP, which resulted in the suppression of HemEC proliferation [27]. Metformin, the most commonly prescribed type 2 diabetes treatment, has been demonstrated to decrease tumor growth by reducing Glut1 expression [56,57,58]. Minoru Ono reported that a patient with a hepatic hemangioma who received treatment with the antidiabetic medication metformin experienced complete remission of the tumor [59]. In this study, we discovered that metformin may have some effect on IH, but unfortunately only at larger doses.

## 5. Conclusions

A high-throughput CD31+ HemEC 3D microtumor model with a uniform size was constructed in this study using vascular-specific dECM and a micropattern array. We used propranolol to test the model’s applicability. We also found that metformin may have some effect on IH. The CD31+ hemangioma endothelial cell-based three-dimensional microtumor models may be effective experimental models for IH mechanistic investigation and drug screening.

## Figures and Tables

**Figure 1 pharmaceuticals-15-01393-f001:**
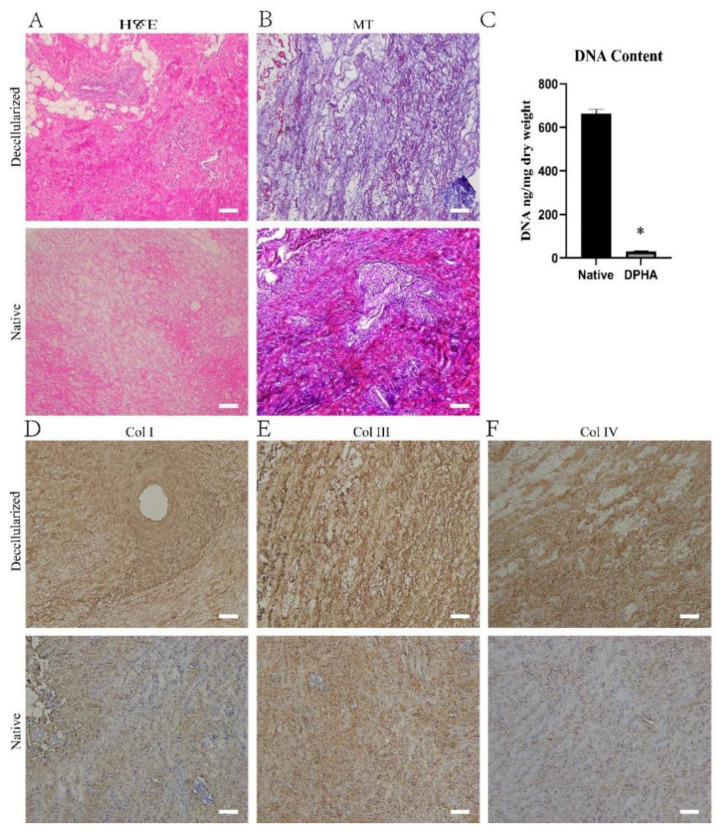
Characteristics of the decellularized porcine heart aortic scaffolds. (**A**) Hematoxylin and eosin staining and (**B**) Masson’s trichrome (MT) staining of native and decellularized porcine heart aortic scaffolds. (**C**) Relative DNA content in native and decellularized porcine heart aortic scaffolds. (**D**) Collagen I, (**E**) collagen Ⅲ, and (**F**) collagen IV staining of native and decellularized porcine heart aortic scaffolds. * *p* < 0.05 versus the native group. Scale bar = 100 μm.

**Figure 2 pharmaceuticals-15-01393-f002:**
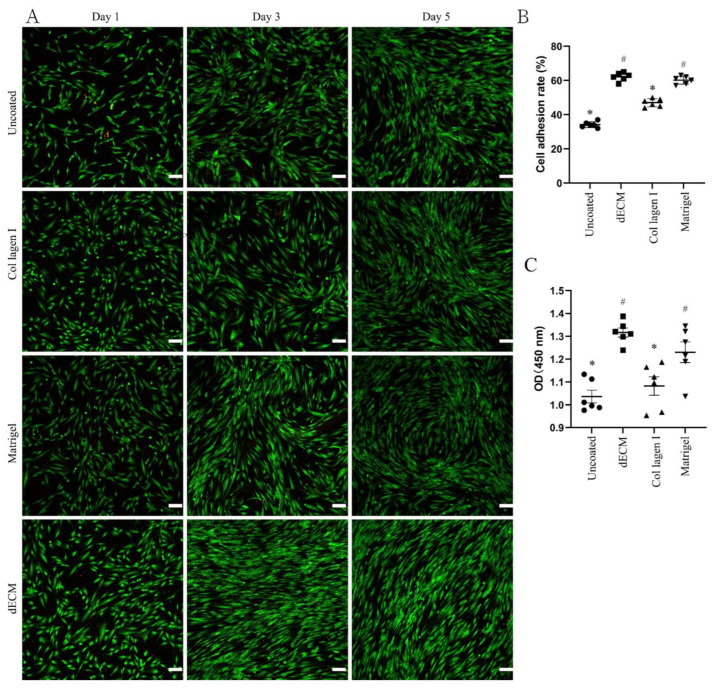
Characteristics of the decellularized extracellular matrix hydrogel derived from cultures of decellularized porcine heart aortic scaffolds. Live/dead staining of (**A**) CD31+ HemECs cultured on different hydrogels on days 1, 3, and 5. The cell adhesion rate of (**B**) CD31+ HemECs on different hydrogels at 10 min after cell seeding. The proliferation rate of (**C**) CD31+ HemECs on different hydrogels at 24 h after cell seeding. * *p* < 0.05 versus the ECM group; # *p* < 0.05 versus the untreated group. Scale bar = 100 μm.

**Figure 3 pharmaceuticals-15-01393-f003:**
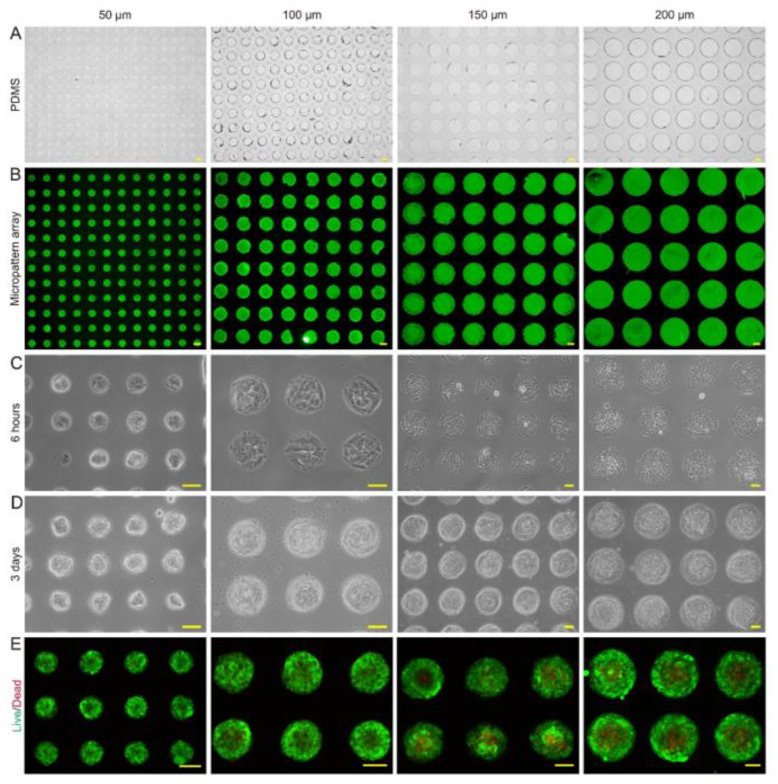
Fabrication of micropattern arrays and the formation of CD31+ HemEC microtumors. Micropattern arrays (**A**) in bright field and (**B**) in fluorescence light from PDMS seals with 50, 100, 150, and 200 μm diameters. (**C**) The CD31+ HemECs cultured on micropattern arrays with different diameters at 6 h after cell seeding. (**D**) The CD31+ HemECs formed microtumors with different diameters at day 3 after cell seeding. (**E**) Live/dead staining of CD31+ HEC microtumors. Scale bars = 50 μm.

**Figure 4 pharmaceuticals-15-01393-f004:**
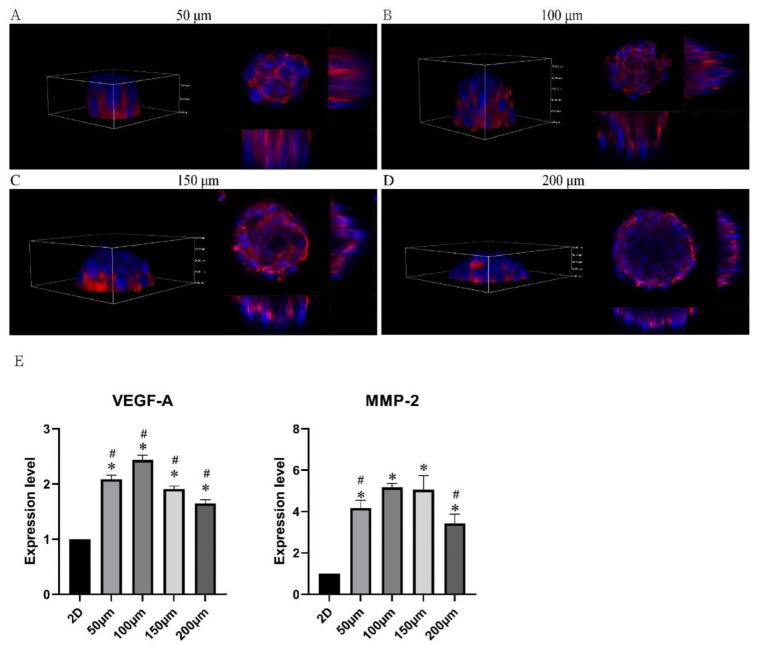
Characteristics of the CD31+ HemEC microtumors. (**A**–**D**) Three-dimensional view of CD31+ HemEC microtumors in micropattern arrays with 50, 100, 150, and 200 μm diameters by confocal microscopy. (**E**) Gene expression levels of MMP-2 and VEGF-A in CD31+ HemEC microtumors with different diameters. * *p* < 0.05 compared to the 2D culture; # *p* < 0.05 compared to the 100 μm diameters.

**Figure 5 pharmaceuticals-15-01393-f005:**
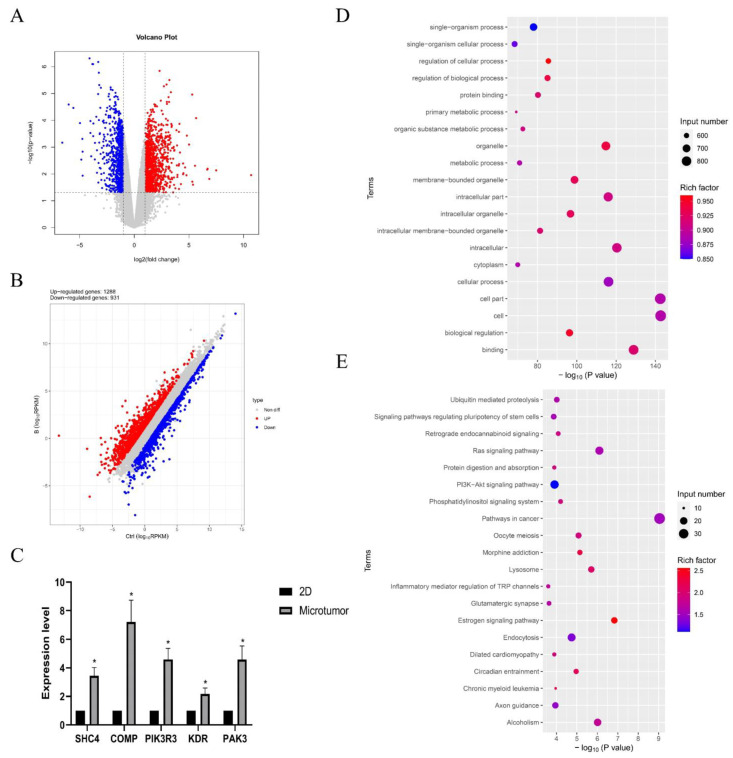
Results of the RNA sequencing. (**A**) Volcano plot; (**B**) the differential expressions of genes between the CD31+ HemEC microtumors and the 2D cell model. High relative expression is indicated by red, whereas low relative expression is indicated by blue. (**C**) Gene expression levels of focal adhesion-related genes (SHC4, COMP, PIK3R3, KDR, and PAK3). (**D**,**E**) The top 20 enriched GO and KEGG signaling pathways for significantly upregulated genes. * *p* < 0.05 compared to the 2D culture.

**Figure 6 pharmaceuticals-15-01393-f006:**
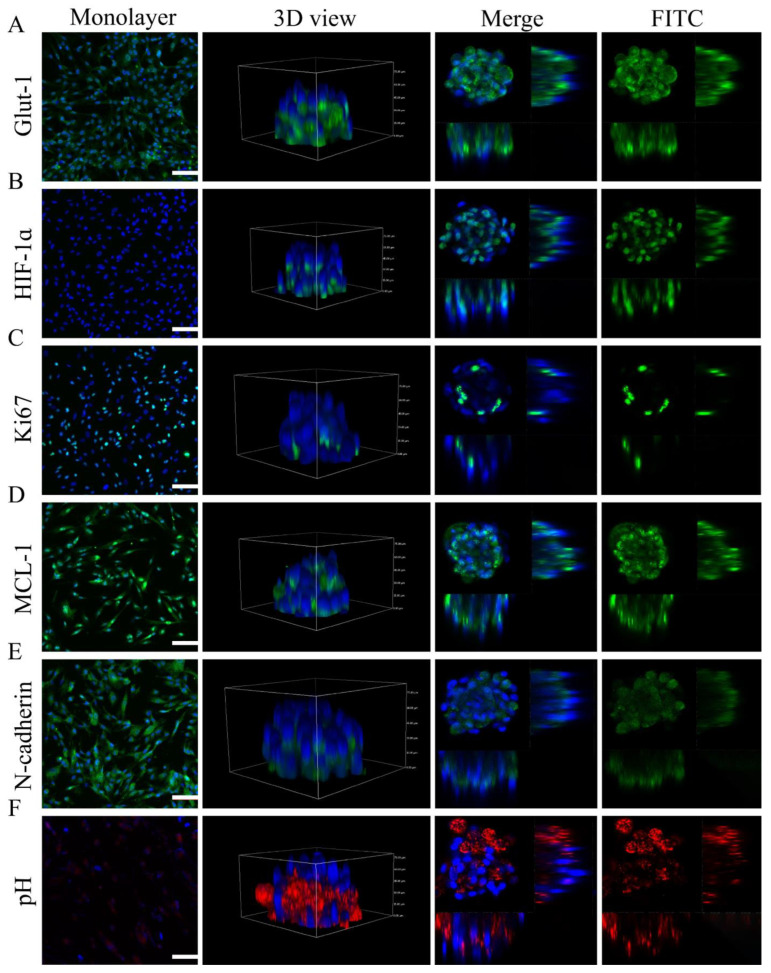
Features of microtumors detected by immunocytochemistry of (**A**) GLUT-1 (specific immunity indicator of IH), (**B**) HIF-1α (hypoxia marker), (**C**) Ki67 (proliferation marker), and (**D**) MCL-1 (antiapoptotic marker) in CD31+ HemEC microtumors and monolayers. Immunocytochemistry of (**E**) N-cadherin in CD31+ HEC microtumors and monolayers. (**F**) Cytosolic pH of CD31+ HemEC microtumors and monolayers. Scale bars = 100 μm.

**Figure 7 pharmaceuticals-15-01393-f007:**
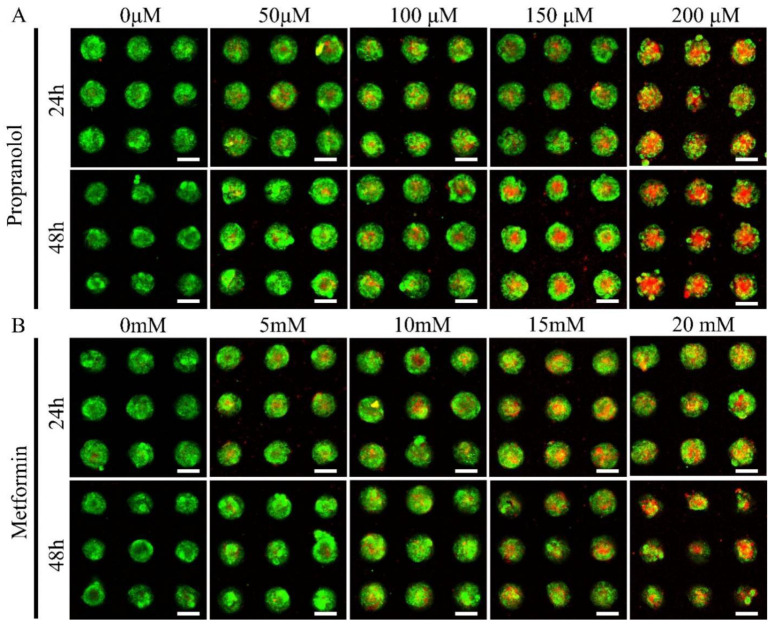
Drug screening of propranolol (**A**) and metformin (**B**) in CD31+ HemEC microtumors with live/dead staining. Scale bars = 100 μm.

## Data Availability

Data is contained within the article and Appendix A.

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
