# Peer review of "Three-Dimensional Microtumor Formation of Infantile Hemangioma-Derived Endothelial Cells for Mechanistic Exploration and Drug Screening"

_pharmaceuticals, 2022, doi:10.3390/ph15111393_

Round 1

Reviewer 1 Report

General comments

This paper aims to develop a more biological relevant 3D in vitro model for infantile hemangioma. The authors should be commended for attempting to develop a unique model system for this disease that could be used in future drug discovery programs. Both quantitative and qualitative assessments have been presenting with optical microscopy used for much of the in vitro microtumour characterisation as well transcriptome analysis. Unfortunately, there is a lack of clear separation between visual assessment of microscopy images and unbiased quantitative image analysis with much of the imaging data presented. The gene expression work is compared to existing monolayer cultures but no correlation with in vivo tumour expression was performed which limits the interpretation of this model to just ‘different’ not ‘more relevant’ expression perhaps.

The authors also target this model for future drug screening high throughput studies. However, I believe a much more in-depth amenability to compound screening study would need to be performed before that conclusion could be recached. For a paper that is assessing drug effects on a tumour model there is also a lack of standard efficacy measures such as dose response curves/Ic50 and quality control assay measures such CV% and technical/biological replicate data. As microscopy is used for much of the 3D culture characterisation, I think a much more in depth methods section of the setup of the imaging protocols is required. Imaging spheroids requires specific optimisation and validation compared to monolayer or thin sample microscopy and this should be highlighted at least in the sup section.  Finally, although much of the paper is written to a high standard there are several examples of spelling mistakes, incorrect tense and punctuation that needs to be addressed for acceptance.

Specific comments

Please check consistency of spelling, tense and punctuation (US language, i.e. ‘obtaine’ line 22) throughout manuscript

Line 25 - Please reword this sentence to make it a standalone statement.

Line 46 – RAS signalling pathway or proteins?

Line 94  - You mention ‘high throughput’ model, in drug screening this would normally be defined as thousands to millions of compounds/conditions in an automated or semi-automated fashion. Can you elaborate on the ability to scale up this micropattern model to this sort of drug discovery standard or is this a proof-of-concept study?

Line 113 – How many tissue sections were used for quantification in Figure 1C?

Line 174 – Which Nikon confocal was used and which settings (objective, laser powers, z-stack step dimensions etc). How was thick sample imaging validated?

Line 200 – ‘immunocytochemistry’ instead of ‘immunofluorescent staining’?

Line 203 – All antibodies are used at very high levels for ICC (1:100), have you previously optimised these antibodies for specificity and confocal microscopy?

Line 208 &213 – How was the ICC work evaluated? Which microscope and what settings?

Line 220 – What was the reason for using individual T-tests rather than ANOVA for some of the multiple stats tests?

Line 249 – Are the individual data points in Figure 2B technical or biological replicates?

Line 54 – As you previously saw differential adhesion rates across different plate coatings, did you normalise your proliferation data to account for the different adhesion rates or just seeing the result of different starting cell populations?

Line 256 – Are the statements of higher cell density a quantitative assessment if so, has confluency been calculated and can be graphed? As the starting cell number is higher in the dECM and Matrgiel coatings they are more likely to reach confluency by the longer time points, is this likely to explain the homogenous appearance of cells. Also can this be quantified by high content imaging perhaps using a shape or texture feature if it is going to be mentioned in the results?  

Line 261  - Should the untreated group be labelled as ‘uncoated’ not ‘uncoating’?

Line 261 – Figure 2 has red scale bars which are quite difficult to see, can these be changed to White?

Line 269 – You state PDMS reliably produce different sizes, do you have any quantification of this assessment (i.e. SD of multiple arrays)?

Line 273 – I’m not sure what the biological meaning of ‘orderly microtumours’ means? Can you relate this to a shape measurement like sphericity or diameter or cell junctions etc?

Line 276 – Without knowing how this was imaged or quantified it is difficult to know what the phrase ‘nearly all cells displayed green fluorescence’ or is this simply a visual assessment. Was a confocal microscope used or widefield, have you accounted for background fluorescence values. It would be great to have some quantitative data with error bars here on consistency of growth across multiple replicates.

Line 278 – More details on the imaging method is required for interpreting these images. From the 3D renders provided it appears the smaller spheroids were imaged to about 50% of their volume and show a spherical formation as you would expect. After ~50um you will reach the limit of confocal microscopy to penetrate thicker samples and clearing reagents/higher permeabilization of dyes will be required to assess volumetric structure. It appears the larger 150 and 200um spheroid renders are not imaged correctly and are just showing imaging artefacts (hollow rings) of thick sample surfaces.

Line 283 – I’m not sure how this conclusion was achieved? Is it simply based on the largest increase in expression of the two genes selected, what is the biological significance of this expression? Shouldn’t the statistical assessment be carried out for all expression levels against each other not just compared to the 2D levels.

Line 286 – The transcriptome sequencing is performed against only 2D cultures not patient derived tumours so is the 3D data just different to monolayer cultures or is closer to the in vivo microtumours?

Line 317  - The language used here of ‘increased expression’ or ‘lower intensity’ indicates some sort of quantification yet there is no graphing of this data just representative images and 3D renders. I also think it is difficult to assess these images as stated previously, it looks like imaging is confined to only the outer half shell of the microtumours as there is limited clearing protocols or reagent permeabilization used for the fluorescent imaging.

Line 335 – What is the biological relevance of the statement ‘good shape and activity’

Line 338 – The 50% lethal dose would normally be calculated as an IC50 molarity using a generated dose response curve in drug screening projects. Please generate curves and IC50 values if you would like to state 50% lethal dosages. Also is this data of 3 biological replicates or is the 3 replicates seen in the images the technical from one assay?

Line 339 – How was the 50% value at 100um calculated – was this image based analysis or just visual qualitative assessment?

Line 347 – Scale bars are now yellow, before they were red please make consistent across all figures

Line 341 – Again an assessment of cell health has been performed here ‘almost all cells dies’. How was this calculated what was the error over multiple biological replicates?

Line344 – Drug concentrations of 5mM or 20mM are extremely high and never usually seen in tumour compound screening assays. Could these values/calculations be checked for accuracy?

Line 348 – Figure legend does not indicate what colours are associated with what dyes for the live/dead staining

Line 353 – As previously stated a high throughput drug screening model will require indications of scalability, amenability to automated liquid handling, robustness over multiple replicates, consistency of reference drugs etc. I think you will need to show these things before you can claim you have developed a model suitable for high throughput drug work.

Line 358 – So what does that tell you about the model you have developed, does it confirm biological or clinical relevance?  The front line treatment for IH required extremely high dosing levels of ~100um to show some cytotoxicity. Is this what is seen in in vivo?

Line 360 – This is interesting insight into the in vivo growth characteristics of IH but how does that relate to your mono-cell type 3D short term culture model you have created? Does your system have different proliferative phases that can be assessed with different treatments?

Line 406 – So dECM was statistically superior to something like Matrigel in all of those categories listed? I did not see this stats comparison in results.

Line 410 – As previously – I disagree with this statement, you showed high expression levels of two genes to come to this conclusion only?

Line 447  - As previously your confocal imaging only showed the outer shell of the microtumours so difficult to make conclusions about the inner core? Need to asses limitations of your imaging.

Line 461 – What does ‘good cell adhesion’ mean?

Line 476 – I would have a different conclusion when a compound is required at a 20mM dose for cytotoxicity effect.

Supp Material

- Spelling ‘Group’

- Please list n values for the data and if bar graph is the mean and SD error bars?

Author Response

Dear Reviewer

Thank you for your prompt attention to our manuscript and thanks a lot for your helpful suggestions. We are pleased to follow your criticism and the manuscript has been revised according to your suggestions.

General comments

  1. This paper aims to develop a more biological relevant 3D in vitro model for infantile hemangioma. The authors should be commended for attempting to develop a unique model system for this disease that could be used in future drug discovery programs. Both quantitative and qualitative assessments have been presenting with optical microscopy used for much of the in vitro microtumour characterisation as well transcriptome analysis. Unfortunately, there is a lack of clear separation between visual assessment of microscopy images and unbiased quantitative image analysis with much of the imaging data presented. The gene expression work is compared to existing monolayer cultures but no correlation with in vivo tumour expression was performed which limits the interpretation of this model to just ‘different’ not ‘more relevant’ expression perhaps.

Response: Thanks for your valuable comment. The fluorescence intensity has measured as a mean gray value and analyzed by ImageJ software for the quantification of the viability of microtumor. It is an excellent idea to compare 3D and monolayer cultures with in vivo tumors using transcriptome sequencing.In this paper, we were focused on the in vitro tumor model. Therefor, the 3D microtumore were mainly compared to 2D. However, compared to the vivo tumor sample, the analysis of transcriptome sequencing is very interesting and will be further explored.

  1. The authors also target this model for future drug screening high throughput studies. However, I believe a much more in-depth amenability to compound screening study would need to be performed before that conclusion could be recached. For a paper that is assessing drug effects on a tumour model there is also a lack of standard efficacy measures such as dose response curves/Ic50 and quality control assay measures such CV% and technical/biological replicate data. As microscopy is used for much of the 3D culture characterisation, I think a much more in depth methods section of the setup of the imaging protocols is required. Imaging spheroids requires specific optimisation and validation compared to monolayer or thin sample microscopy and this should be highlighted at least in the sup section.  Finally, although much of the paper is written to a high standard there are several examples of spelling mistakes, incorrect tense and punctuation that needs to be addressed for acceptance.

Response: Thanks for your comment.In addition, we have checked this paper repeatedly to reduce the grammar mistakes and inaccurate description as far as possible. The corresponding explanation of each point which is raised in your comments as follows:

Specific comments

  1. Please check consistency of spelling, tense and punctuation (US language, i.e. ‘obtaine’ line 22) throughout manuscript

Response: Thanks for your correction. The consistency of spelling, tense and punctuation have been corrected in the manuscript.

  1. Line 25 - Please reword this sentence to make it a standalone statement.

Response: Thanks for your correction. This sentence has been corrected in the manuscript.

  1. Line 46 – RAS signalling pathway or proteins?

Response: Thanks for your correction. RAS means renin-angiotensin system. The relevant content has been added in the manuscript.

  1. Line 94  - You mention ‘high throughput’ model, in drug screening this would normally be defined as thousands to millions of compounds/conditions in an automated or semi-automated fashion. Can you elaborate on the ability to scale up this micropattern model to this sort of drug discovery standard or is this a proof-of-concept study?

Response: Thank you for your suggestion. Micropattern arrays can effectively modulate the concordance of spheroids’ morphological characteristics, such as shape, size and arrangement, which is highly advantageous for high-throughput system-based drug screening combining with confocal microscopy analyzing fluorescence intensity. This system eliminates the need for extracting the formed tumor spheroids and then performing the analysis in a separate well plate. Therefore, this system is very promising for high-throughput drug screening.

  1. Line 113 – How many tissue sections were used for quantification in Figure 1C?

Response: Thank you for your suggestion. Respectively, five independent fresh and decellularized porcine heart aortic tissue were used for DNA quantification. The relevant content has been added in the manuscript.

  1. Line 174 – Which Nikon confocal was used and which settings (objective, laser powers, z-stack step dimensionsetc). How was thick sample imaging validated?

Response: Thank you for your suggestion. The objective magnification was ×10. The laser power of 405, 488 and 561nm were 2, 5 and 2%, respectively. The z-stack step was 2 μm.

  1. Line 200 – ‘immunocytochemistry’ instead of ‘immunofluorescent staining’?

Response: Thanks for your correction. This related content has been corrected in the manuscript.

  1. Line 203 – All antibodies are used at very high levels for ICC (1:100), have you previously optimised these antibodies for specificity and confocal microscopy?

Response: Thank you for your suggestion. The concentration of all antibodies were chosen as recommended by the manufacturers instructions (abcam).

  1. Line 208 &213 – How was the ICC work evaluated? Which microscope and what settings?

Response: Thank you for your suggestion. Several characteristics of microtumor (cell adhesion, hypoxia, proliferation etc.) were determined by Immunofluorescence Staining. Then, Fluorescent images were acquired with a observed by two-photon confocal microscope (A1RMP+, Nikon, Japan). The objective magnification was ×10. The z-stack step was 2 μm.The objective magnification was ×10. The laser power of 405, 488 and 561nm were 2, 5 and 2%, respectively. The z-stack step was 2 μm.

  1. Line 220 – What was the reason for using individual T-tests rather than ANOVA for some of the multiple stats tests?

Response: Thank you for your suggestion. T-test was used to compare two groups, while ANOVA was used to compare three or more groups. This related content has been added to methodology in the manuscript.

  1. Line 249 – Are the individual data points in Figure 2B technical or biological replicates?

Response: Thank you for your suggestion. The individual data points was the biological replicates in the Figure 2B.

  1. Line 54 – As you previously saw differential adhesion rates across different plate coatings, did you normalise your proliferation data to account for the different adhesion rates or just seeing the result of different starting cell populations?

Response: Thank you for your suggestion. The number of seeding cell was same. After 2 h of seeding, the cell adhesion rate was evaluated using Countess Ⅱ FL (Invitrogen, USA) counting of non‐adhered cells. The effect of cell proliferation was ignored, because of the short adhesion time.

  1. Line 256 – Are the statements of higher cell density a quantitative assessment if so, has confluency been calculated and can be graphed? As the starting cell number is higher in the dECM and Matrgiel coatings they are more likely to reach confluency by the longer time points, is this likely to explain the homogenous appearance of cells. Also can this be quantified by high content imaging perhaps using a shape or texture feature if it is going to be mentioned in the results?  

Response: Thank you for your suggestion. On the first day of culture, the number of CD31+ HemECs was calculated by ImageJ software on coated and uncoated substrate. This related graph has been added in the manuscript. We totally agree with your viewpoint. The number of CD31+ HemECs and the result of CCK-8 indicated that the proliferation of CD31+ HemECs were higher on the dECM and Matrgiel than collagen I and uncoated, which leaded the number of CD31+ HemECs was higher on the dECM and Matrgiel than collagen I and uncoated on the 3 and 5 days, is this likely to explain the homogenous appearance of cells. The homogenous phenomenon was estimated by observing the Morphology of CD31+ HemECs.

  1. Line 261  - Should the untreated group be labelled as ‘uncoated’ not ‘uncoating’?

Response: Thanks for your correction. This related content has been corrected in the manuscript.

  1. Line 261 – Figure 2 has red scale bars which are quite difficult to see, can these be changed to White?

Response: Thanks for your correction. The color of scale bars has been changed in the Figure 2.

  1. Line 269 – You state PDMS reliably produce different sizes, do you have any quantification of this assessment (i.e. SD of multiple arrays)?

Response: Thank you for your suggestion. The parameters of micropattern arrays were designed by ourselves. The PDMS seals were fabricated by Suzhou CChip scientific instrument Co., Ltd. Therefor, the different sizes of PDMS seals could be reliably fabricated. The related content has been added in the manuscript.

  1. Line 273 – I’m not sure what the biological meaning of ‘orderly microtumours’ means? Can you relate this to a shape measurement like sphericity or diameter or cell junctions etc?

Response: Thank you for your suggestion. Different multicellular spheroid techniques (such as hanging drop, microwell technique and rocker system) often produce spheroids of different size and shape, which can strongly influence the outcome of drug delivery and efficacy. Consequently, it is necessary to modulate the concordance of spheroids’ morphological characteristics to reducing error and increasing reproducibility. Micropattern arrays can effectively modulate the the concordance of spheroids’ morphological characteristics, such as shape, size and arrangement, which is highly advantageous for high-throughput system-based drug screening. Consequently, the consistency of size and arrangement of microtumors were evaluated.

  1. Line 276 – Without knowing how this was imaged or quantified it is difficult to know what the phrase ‘nearly all cells displayed green fluorescence’ or is this simply a visual assessment. Was a confocal microscope used or widefield, have you accounted for background fluorescence values. It would be great to have some quantitative data with error bars here on consistency of growth across multiple replicates.

Response: Thank you for your suggestion. The viability of microtumore was evaluated by FluoroQuench fluorescent staining. Fluorescent images were acquired with a confocal microscope, the fluorescence intensity measured as a mean gray value and analyzed by ImageJ software for the quantification of the viability of microtumor. The related content has been added in the manuscript.

  1. Line 278 – More details on the imaging method is required for interpreting these images. From the 3D renders provided it appears the smaller spheroids were imaged to about 50% of their volume and show a spherical formation as you would expect. After ~50um you will reach the limit of confocal microscopy to penetrate thicker samples and clearing reagents/higher permeabilization of dyes will be required to assess volumetric structure. It appears the larger 150 and 200um spheroid renders are not imaged correctly and are just showing imaging artefacts (hollow rings) of thick sample surfaces.

Response: Thank you for your suggestion. The volumetric structure of microtumor was evaluated by two-photon confocal microscope using water lens, which could be effectively evaluated the 3D architecture of microtumor analyzed by studying the Z-axis. Because of cells were restrictively adhered to the micropattern, resulting in the limited cell growth. Furthermore, cells will spontaneously assemble into microtumore with 3D multicellular structure by proliferation of cells and cell-cell adhesion abilities. Therefore, the bottom of microtumor was flat and the feature of microtumor was half shell.

  1. Line 283 – I’m not sure how this conclusion was achieved? Is it simply based on the largest increase in expression of the two genes selected, what is the biological significance of this expression? Shouldn’t the statistical assessment be carried out for all expression levels against each other not just compared to the 2D levels.

Response: Thank you for your suggestion. The consistency of size and arrangement, viability, 3D architecture and several gene expression of mircotumor was further evaluated for screening the optical size of micropattern. In regard to the expression of the genes, we added another analysis, “compared to 100 μm”.The related content has been added in the manuscript. 

  1. Line 286 – The transcriptome sequencing is performed against only 2D cultures not patient derived tumours so is the 3D data just different to monolayer cultures or is closer to the in vivo microtumours?

Response: Thank you for your suggestion. Generally, standard preclinical screening procedures for anticancer agents involve target identification of a compound on immortalized cell lines cultured in 2D. In this paper, we were focused on the in vitro tumor model. Therefor, the 3D microtumore were mainly compared to 2D. However, compared to the vivo tumor sample, the analysis of transcriptome sequencing is very   interesting and will be next explored.

  1. Line 317  - The language used here of ‘increased expression’ or ‘lower intensity’ indicates some sort of quantification yet there is no graphing of this data just representative images and 3D renders. I also think it is difficult to assess these images as stated previously, it looks like imaging is confined to only the outer half shell of the microtumours as there is limited clearing protocols or reagent permeabilization used for the fluorescent imaging.

Response: Thank you for your suggestion. Special attention has been paid to several spatial characteristics of microtumor (such as cell proliferation, hypoxia microenvironment, antiapoptotic ability and acid microenvironment). Therefor, these spatial characteristics of microtumor were qualitatively  analysed. We totally agree with your viewpoint. The permeabilization is very important to immunofluorescent staining of overall tissue. Through two-photon confocal microscope using water lens, the volumetric structure of microtumor could be observed. However, the limitations of confocal microscope had to be considered, which was very interesting and will be next explored in the future.

  1. Line 335 – What is the biological relevance of the statement ‘good shape and activity’

Response: Thank you for your suggestion. After 3 days culture, the CD31+ HemEC microtumors possessed some characteristics, such as the multicellular structure, proliferative zone of outer layer of microtumors, hypoxic inner region of microtumors, the acidic microenvironment, etc. Meanwhile, the microtumors showed the good cellular viability.

  1. Line 338 – The 50% lethal dose would normally be calculated as an IC50 molarity using a generated dose response curve in drug screening projects. Please generate curves and IC50 values if you would like to state 50% lethal dosages. Also is this data of 3 biological replicates or is the 3 replicates seen in the images the technical from one assay?

Response: Thank you for your suggestion. I apologize for bothering you due of my sloppy description.We have revised this section of the description because it is not clear enough. “The cell viability rate was approximately 52.9% when the propranolol dosage was 100μM.at the 2D culture level, according to CCK-8 analysis (Figure S1). The cell viabilities of propranolol were 71.3±4.5%, 64.7±2.3%, 54.1± 1.6% and 46.7 ±2.3% , respectively, at 50 μM, 100 μM, 150 μM and 200 μM at 24h, while the cell viabilities were 62.4±0.5%, 54.2%±0.4, 49.7± 1.5% and 39.3 ±1.2% at 48h after drug administration (Figure 7A). The cytotoxic effect of metformin was lowe , the cell viabilities of metformin were 72.1±1.1%, 61.3±4.4, 50.6% ± 0.8% and 45.0 ±1.9%, respectively, at 5 mM, 10 mM, 15 mM, 20 mM at 24h, while the cell viabilities were 73.1±2.8% , 55.7±0.6%, 45.5 ± 2.8 and 40.8 ±2.0 % at 48h after drug administration (Figure 7B).”

  1. Line 339 – How was the 50% value at 100um calculated – was this image based analysis or just visual qualitative assessment?

Response:Thank you for your suggestion. I apologize again for bothering you due of my sloppy description.We have revised this section of the description because it is not clear enough.

  1. Line 347 – Scale bars are now yellow, before they were red please make consistent across all figures

Response:Thank you for your suggestion. We try my best to unite the bar into white, but the yellow bar is still used because Figure 3 contains some photos with a white background.

  1. Line 341 – Again an assessment of cell health has been performed here ‘almost all cells dies’. How was this calculated what was the error over multiple biological replicates?

Response:Thank you for your suggestion. We have revised this section of the description because it is not clear enough.

  1. Line344 – Drug concentrations of 5mM or 20mM are extremely high and never usually seen in tumour compound screening assays. Could these values/calculations be checked for accuracy?

Response: Thank you for your suggestion. Metformin hydrochloride ( inhibits proliferation of ESCs in a concentration-dependent manner. The IC50 is 2.45 mM for A-ESCs and 7.87 mM for N-ESCs. In this study, it is disappointing that metformin had a considerable effect at concentrations up to 15 mM.

  1. Line 348 – Figure legend does not indicate what colours are associated with what dyes for the live/dead staining

Response: Thank you for your suggestion. The viability of microtumore was evaluated by FluoroQuench fluorescent staining.

  1. Line 353– As previously stated a high throughput drug screening model will require indications of scalability, amenability to automated liquid handling, robustness over multiple replicates, consistency of reference drugs etc. I think you will need to show these things before you can claim you have developed a model suitable for high throughput drug work.

Response: Thank you for your suggestion. The related sentence has been revised. We then successfully constructed a homogenous 3D IH-derived CD31+ HemEC model.

  1. Line 358 – So what does that tell you about the model you have developed, does it confirm biological or clinical relevance?  The front line treatment for IH required extremely high dosing levels of ~100um to show some cytotoxicity. Is this what is seen in in vivo?

Response: Thank you for your suggestion. Currently, the mechanism of propranolol on IH has not been understood, and greater concentrations are frequently required in in vitro models, which will be the subject of future study.

  1. Line 360 – This is interesting insight into the in vivo growth characteristics of IH but how does that relate to your mono-cell type 3D short term culture model you have created? Does your system have different proliferative phases that can be assessed with different treatments?

Response: Thank you for your suggestion. Because this paragraph is somewhat redundant, the trimmed portion has been relocated to the introduction.

  1. Line 406 – So dECM was statistically superior to something like Matrigel in all of those categories listed? I did not see this stats comparison in results.

Response: Thank you for your suggestion. Through analysing the ability of cell adhesion, cell proliferation and homogenous phenomenon, we found that using dECM-based hydrogels as the surface coating matrix was more beneficial to cell adhesion, proliferation, and homogenous phenomenon of CD31+ HemECs than collagen I and uncoated. However, compared with Matrgiel, the difference was not statistically significant. The related content has been revised in the manuscript. 

  1. Line 410 – As previously – I disagree with this statement, you showed high expression levels of two genes to come to this conclusion only?

Response: Thank you for your suggestion. The consistency of size and arrangement, viability, 3D architecture and several gene expression of mircotumor was further evaluated for screening the optical size of micropattern.

  1. Line 447  - As previously your confocal imaging only showed the outer shell of the microtumours so difficult to make conclusions about the inner core? Need to asses limitations of your imaging.

Response: Thank you for your suggestion. We totally agree with your viewpoint. Through two-photon confocal microscope using water lens, the volumetric structure of microtumor could be observed. However, the limitations of confocal microscope had to be considered, which was very interesting and will be next explored in the future. 

  1. Line 461 – What does ‘good cell adhesion’ mean?

Response: Thank you for your suggestion. The CD31+ HemEC microtumors had good cell-cell adhesion. The related content has been revised in the manuscript .

  1. Line 476 – I would have a different conclusion when a compound is required at a 20mM dose for cytotoxicity effect.

Response: Thank you for your suggestion. The sentence has been revised to “We discovered that metformin may has some effect on IH, but unfortunately only at larger doses”.

Reviewer 2 Report

The manuscript is very interesting and well written. The experimental design is clear and gives precise and interesting answers. I have however a few comments:

It is not clear to me why the need of printing pattern array, when first of all, in nature tumours are not uniform and stable. Second, there are commercially available microplates which allow precise reproducibilty of the cultures.

In paragraph 3.6 Drug Screening (line 336), it is not clear how the authors evaluated tumour activity. If it is by marker expression or they used different methods.

I think that a big part of the discussion should be moved to the introduction (i.e. lines 371-389, 422-433 etc).

A few minor comments: 

line 21: I would correct "we try" with "we will try".

line 313: I am pretty sure that the colour that indicates low expression is blue, rather than green. 

All in all it is a very interesting paper and I would be very curious to see if the tumour response to the drugs remains invariable if to the mini-tumours is added a vascular flow. But I am sure this would make another interesting paper in the future.

Author Response

Dear Reviewer

Thank you for your prompt attention to our manuscript and thanks a lot for your helpful suggestions. We are pleased to follow your criticism and the manuscript has been revised according to your suggestions.

  1. It is not clear to me why the need of printing pattern array, when first of all, in nature tumours are not uniform and stable. Second, there are commercially available microplates which allow precise reproducibilty of the cultures.

Response: Thank you for your suggestion. Generally, standard preclinical screening procedures for anticancer agents involve target identification of a compound on immortalized cell lines cultured in 2D. Once a target has been identified, experiments of different complexity are performed using computational, in vitro, and in vivo models. Comparing with 2D culture, 3D spheroids possess many features resembling in vivo tumors including cell-cell and cell-ECM interactions, hypoxia and central necrosis, and drug resistance. Therefore, 3D spheroids are a promising model in vitro. The related content has been added in the manuscript.

Different multicellular spheroid techniques (such as hanging drop, microwell technique and rocker system) often produce spheroids of different size and shape, which can strongly influence the outcome of drug delivery and efficacy. Consequently, it is necessary to modulate the concordance of spheroids’ morphological characteristics to reducing error and increasing reproducibility. Micropattern arrays can effectively modulate the concordance of spheroids’ morphological characteristics, such as shape, size and arrangement, which is highly advantageous for high-throughput system-based drug screening.

  1. In paragraph 3.6 Drug Screening (line 336), it is not clear how the authors evaluated tumour activity. If it is by marker expression or they used different methods.

Response: Thank you for your suggestion. The cytotoxicity of drugs was examined by detecting the viability of CD31+ HemEC microtumors. Next, the viability of microtumors was analyzed by FluoroQuench fluorescent staining. The related content has been added in the manuscript

  1. I think that a big part of the discussion should be moved to the introduction (i.e. lines 371-389, 422-433 etc).

Response: Thank you for your suggestion. We have moved this part of the discussion to the introduction.

A few minor comments: 

  1. line 21: I would correct "we try" with "we will try".

Response: Thanks for your correction. The relevant mistake has been corrected in the manuscript.

  1. line 313: I am pretty sure that the colour that indicates low expression is blue, rather than green. 

Response: Thanks for your correction. The relevant mistake has been corrected in the manuscript.

  1. All in all it is a very interesting paper and I would be very curious to see if the tumour response to the drugs remains invariable if to the mini-tumours is added a vascular flow. But I am sure this would make another interesting paper in the future.

Response: Thank you for your suggestion. This topic is very interesting and will be next explored in the future.

Reviewer 3 Report

The paper titled ‘Three-dimensional microtumor formation of infantile hemangioma-derived endothelial cells for mechanistic exploration and drug screening’ authored by Y. Li et al. reports the fabrication of IH-derived CD31+ hemangioma endothelial cell three-dimensional microtumor models by using a micropattern array obtained with vascular-specific extracellular matrix by decellularization of porcine aorta as the bioink.  

The manuscript is well written and clear. The topic is interesting and the findings are new. I only have a few concerns that need to be addressed before that the manuscript can be accepted for publication in Pharmaceuticals. A list of comments can be found below:

-       Unless requested by the journal, I suggest the authors to remove the subtitles in the abstract (introduction, objective, results and conclusion). I would rather add a paragraph of Conclusion at the end of the work where the findings of the work are well summarized.

-       The purpose of the experiments is not always clear. I suggest the the author to add a small sentence at the beginning of each paragraph of the results to explain the goal of that specific experiments or measurement. 

-       All the acronyms need to be explained the first time they are mentioned. 

-       The scale bar is not clear in Figure 2.

-       Figure 7B. How can the authors explain that when administering Metfornin at 5 and 10mM the number of the dead cells of the necrotic core seems to decrease after 48 h? 

Author Response

Dear Reviewer

Thank you for your prompt attention to our manuscript and thanks a lot for your helpful suggestions. We are pleased to follow your criticism and the manuscript has been revised according to your suggestions.

  1. Unless requested by the journal, I suggest the authors to remove the subtitles in the abstract (introduction, objective, results and conclusion). I would rather add a paragraph of Conclusion at the end of the work where the findings of the work are well summarized.

Response: Thank you for your suggestion. We have removed the subtitles in the abstract (introduction, objective, results and conclusion).

  1. The purpose of the experiments is not always clear. I suggest the the author to add a small sentence at the beginning of each paragraph of the results to explain the goal of that specific experiments or measurement. 

Response: Thank you for your suggestion. The relevant content has been added in the manuscript.

  1. All the acronyms need to be explained the first time they are mentioned. 

Response: Thanks for your correction. The relevant content has been added in the manuscript.

  1. The scale bar is not clear in Figure 2.

Response: Thanks for your correction. The scale bar was corrected in Figure 2.

  1. Figure 7B. How can the authors explain that when administering Metfornin at 5 and 10mM the number of the dead cells of the necrotic core seems to decrease after 48 h? 

Response: Thank you for your suggestion. Actually, cell viability was slightly lower after 24 hours than after 48 hours (72.1±1.1%and 73.1±2.8%, respectively, measured as a mean gray value and analyzed by ImageJ software), but the difference was not statistically significant. These data were not shown in the manuscript.

Round 2

Reviewer 1 Report

Thank you for addressing the previous concerns. As mentioned below in the individual comments, please provide more of the imaging equipment details in the methods sections and also include the image analysis methods. 

Imaging methods section

Please add the below ImageJ quantification comment to methods section as it is a unique image analysis protocol for this particular model and not just a standard vendor method.

Response: Thanks for your valuable comment. The fluorescence intensity has measured as a mean gray value and analyzed by ImageJ software for the quantification of the viability of microtumor.

Line 194

Please add below response to methods section and include Nikon confocal model. Also was a maximum projection image produced or was analysis performed on the 3D z-stack data?

Response: Thank you for your suggestion. The objective magnification was ×10. The laser power of 405, 488 and 561nm were 2, 5 and 2, respectively. The z-stack step was 2 μm.

Line 218

Please add below response to appropriate methods section and as before include image analysis method such as max projection or 3d dataset used for the characterisation studies.

Response: Thank you for your suggestion. Several characteristics of microtumor (cell adhesion, hypoxia, proliferation etc.) were determined by Immunofluorescence Staining. Then, Fluorescent images were acquired with a observed by two-photon confocal microscope (A1RMP+, Nikon, Japan). The objective magnification was ×10. The z-stack step was 2 μm.The objective magnification was ×10. The laser power of 405, 488 and 561nm were 2, 5 and 2, respectively. The z-stack step was 2 μm.

Line 273

Sorry I don’t understand this response:

Response: Thank you for your suggestion. The number of seeding cell was same. After 2 h of seeding, the cell adhesion rate was evaluated using Countess FL (Invitrogen, USA) counting of non‐adhered cells. The effect of cell proliferation was ignored, because of the short adhesion time.

“You have mentioned ‘Compared to the untreated and collagen I-coated plates, the dECM-coated and Matrigel-coated plates exhibited superior cell proliferation rates (Figure 2C).” However, the sentence before you mention you had different adhesion rates across the treatment groups. Therefore, you actually have different starting cell numbers (that attached). Are you measuring proliferation rate differences (Fig 2C) or simply that you started with less cells attached in some conditions (untreated) and so they had a lower cell count to start with and ultimately lower ‘proliferation’ reading at the 24h timepoint when you ran the CCk-8 kit?

Line 282

Please add the n values to Figure 2 legend.

Response: Thank you for your suggestion. The individual data points was the biological replicates in the Figure 2B.

Line 333

Please add statistical info to his figure legend as you have a bar graph in 5C (n values, stats legend, mean and error bar description)

Line 369

Spelling ‘Lowe’

‘the cell viabilities of metformin…’ This phrase needs to be reworded as it is not the viability of the drug, it is the viability of the cells exposed to that particular drug.

Line 372

Please indicate n values (n=3?) for this quantification somewhere in the text.

As mentioned previously most assay development studies would calculate IC50 values (in R or Graphpad etc) for reference drugs. As you only have 5 data points per drug this would be difficult to calculate for this study but in the future, calculating IC50 values using a 6-12 dose response curve would give much more useful data to the reader on reproducibility and efficacy in the models you develop.

Figure s2

As previously spelling ‘Groupe’ to ‘Group’

Please list n values for the data and if bar graph is the mean and SD error bars?